REGISTERED REPORT PROTOCOL

# Prediction and diagnosis of chronic kidney disease development and progression using machine-learning: Protocol for a systematic review and meta-analysis of reporting standards and model performance

**Fangyue Chen[1,2,3], Piyawat Kantagowit[3], Tanawin Nopsopon[3,4], Arisa Chuklin[3], Krit Pongpirul[3,5,6]** *

**1** Global Health Partnerships, Health Education England, London, United Kingdom, **2** School of Public Health, Faculty of Medicine, Imperial College London, London, United Kingdom, **3** Faculty of Medicine, Chulalongkorn University, Bangkok, Thailand, **4** Harvard T.H. Chan School of Public Health, Boston, Massachusetts, United States of America, **5** Department of International Health, Johns Hopkins Bloomberg School of Public Health, Baltimore, Maryland, United States of America, **6** Clinical Research Center, Bumrungrad International Hospital, Bangkok, Thailand

* krit.po@chula.ac.th, doctorkrit@gmail.com

## Abstract

Chronic Kidney disease (CKD) is an important yet under-recognized contributor to morbidity and mortality globally. Machine-learning (ML) based decision support tools have been developed across many aspects of CKD care. Notably, algorithms developed in the prediction and diagnosis of CKD development and progression may help to facilitate early disease prevention, assist with early planning of renal replacement therapy, and offer potential clinical and economic benefits to patients and health systems. Clinical implementation can be affected by the uncertainty surrounding the methodological rigor and performance of ML-based models. This systematic review aims to evaluate the application of prognostic and diagnostic ML tools in CKD development and progression. The protocol has been prepared using the Preferred Items for Systematic Review and Meta-analysis Protocols (PRISMA-P) guidelines. The systematic review protocol for CKD prediction and diagnosis have been registered with the International Prospective Register of Systematic Reviews (PROSPERO) (CRD42022356704, CRD42022372378). A systematic search will be undertaken of PubMed, Embase, the Cochrane Central Register of Controlled Trials (CENTRAL), the Web of Science, and the IEEE Xplore digital library. Studies in which ML has been applied to predict and diagnose CKD development and progression will be included. The primary outcome will be the comparison of the performance of ML-based models with non-ML-based models. Secondary analysis will consist of model use cases, model construct, and model reporting quality. This systematic review will offer valuable insight into the performance and reporting quality of ML-based models in CKD diagnosis and prediction. This will inform clinicians and technical specialists of the current development of ML in CKD care, as well as direct future model development and standardization.

**Data Availability Statement:** All relevant data from this study will be made available upon study completion.

**Funding:** The authors received no specific funding for this work.

**Competing interests:** The authors have declared that no competing interests exist.

## Introduction

Chronic kidney disease (CKD) is an important non-communicable disease that contributes to significant morbidity and mortality on a global scale, directly or through cardiovascular diseases attributable to impaired kidney function. Its estimated prevalence ranges from 9.1 to 15.1%. It has increased by 29.3% since 1990 due to the increase in the major chronic diseases that contribute to its development, notably diabetes mellitus and hypertension [1, 2]. Nevertheless, CKD is underrecognized by patients, clinicians, and health authorities. The disease often progresses insidiously to end-stage kidney disease (ESKD) with late presentation of symptoms and signs [3, 4]. The life-sustaining treatment for ESKD, renal replacement therapy, poses a significant economic burden on patients and health systems, meaning that currently, an estimated 47 to 73% of individuals are unable to receive it, leaving around 2.3 million individuals dying prematurely [5]. Strategies to prevent or delay CKD onset and progression can potentially lower overall morbidity and mortality while minimizing cost.

Machine Learning (ML), a subset of artificial intelligence (AI), has seen exponential growth across healthcare [6, 7]. ML utilizes a specific dataset to generate an algorithm that employs unknown or varied combinations of complex features and weights to predict the outcome of future inputs [8]. ML-based decision support tools have been developed across many aspects of CKD care across disease prevention, diagnosis, and treatment [7], fuelled by the growth in volume and variety of big data in nephrology and healthcare in general [9, 10]. Notably, algorithms developed in the prediction and diagnosis of CKD development and progression to ESKD may help to facilitate early disease prevention, assist with early care planning, and allocate resources for the most significant clinical benefit [11–15].

Despite the growing promise of ML, several factors can hinder its clinical uptake. These include uncertainty surrounding the performance of ML and the methodological rigor behind its development. Non-ML-based prediction tools for CKD progression and prognosis have been developed and validated, such as the Kidney Failure Risk Equation, which has been used clinically to guide referrals to multidisciplinary CKD clinics [16–18]. Comparisons have been made between ML and non-ML-based prediction tools in general, specifically to chronic diseases and prediction of acute kidney injury, which found similar performance between prediction models developed with ML and conventional logistic regression (LR) techniques [19–21]. In addition, previous studies have questioned the reporting quality and methodology of CKD prediction models [22, 23], as well as other AI-based models in imaging [24], oncology [25], and COVID-19 [26].

This systematic review aims to provide a comprehensive, in-depth summary and evaluation of ML-based diagnostic and prognostic tools for CKD development and progression, which will help to better direct future research strategy and methodology in developing ML algorithms in CKD care.

The proposed systematic review aims to answer the following questions:

1. How do ML-based prediction tools in CKD development and progression perform compared with tools developed using conventional techniques?

2. What are the use cases and constructs of these prediction tools?

3. How are the methodological characteristics and reporting quality of the ML-based tools?

## Materials and methods

The systematic review protocol was registered with the International Prospective Register of Systematic Reviews (PROSPERO) on 26/09/22 for CKD prediction (CRD42022356704) and

CKD diagnosis (CRD42022372378). The protocol followed the Preferred Reporting Items for Systematic Review and Meta-Analysis Protocols (PRISMA-P) 2015 statement [27]. The Checklist for critical Appraisal and data extraction for systematic Reviews of prediction Modelling Studies (CHARMS) has been used to formulate review questions and data extraction [28].

## Study eligibility criteria

**Study designs.** Any peer-reviewed primary studies which assessed a prediction algorithm that utilizes ML techniques applied to clinical problems in the prediction and diagnosis of chronic kidney disease development and progression, including those for CKD screening, CKD prevention, profiling of biomarkers contributing towards CKD, profiling of risk factors leading to CKD, estimation of glomerular filtration rate (GFR) and creatinine levels, prediction of occurrence of CKD, prediction of CKD stages, CKD diagnosis, CKD prognostication, prediction of CKD progression to ESKD and/or requirement for renal replacement therapy, ESKD diagnosis will be included.

Exclusion criteria are 1) Studies that utilize only image-based inputs as the different model development processes require alternative extraction and appraisal tools; 2) studies assessing prediction models of CKD complications other than its progression, including non-exhaustively anemia, electrolyte disturbances, bone disorders, and cardiovascular events; 3) prediction model of RRT including non-exhaustively the choice of RRT modalities which are hemodialysis, peritoneal dialysis, and renal transplantation, and 4) studies reporting only treatment-related outcomes of CKD such as adverse events, rate of complications, the management of complications; 5) informal publication types such as case studies, commentaries, letters to the editor, editorials, meeting abstracts, proceeding papers, conference abstracts, protocols, guidelines, and recommendations; 6) review articles such as narrative review, overview, systematic review, meta-analysis; 7) studies that include participants < 18 years old; 8) animal studies.

**Study participants.** Adult humans whose age was equal to or more than 18 years old.

**Types of interventions.** The studies will present prediction models utilizing ML techniques, including non-exhaustively various regression techniques, decision trees, random forests, support vector machines, K-nearest neighbor, and neural networks, as defined by individual studies. The models will be for the prediction and diagnosis of chronic kidney disease development and progression with or without mention of ESKD.

**Comparators.** We will include studies that compare the performance of ML-based prediction models with those that utilize conventional techniques, including non-exhaustively those that use logistic regression (including penalized LR), cox regression, Poisson regression, least squares linear separation, generalized additive models, discriminant analysis, generalized estimation equations, risk scores, and expert views. Studies that utilize only ML-based tools will also be included.

## Study outcomes

**Primary outcome.**

- Performance comparison of ML-based and non-ML-based prediction tools in CKD development and progression

**Secondary outcomes.**

- ML-based model use case

- Performance of ML-based prediction tool in CKD development and progression

- Stages of model development (internal or external validation or clinical implementation)

- Model development team specialty and the involvement of model end-user such as clinicians during model development

- Evidence of model reporting quality description

- Characteristics of the dataset (size of training, validation and testing datasets, source of dataset, population group, data period, length of follow-up)

- Prediction model construct including ML-based and non-ML based techniques

- Predictor characteristics and selection

- Model outcome characteristics and selection

- Model performance measures used

## Information sources and search strategy

We will search through five databases: PubMed, Embase, the Cochrane Central Register of Controlled Trials (CENTRAL), Web of Science, and the IEEE Xplore digital library. The search strategy is constructed by two health information specialists with systematic review experiences, combining search terms and subject headings (MeSH) related to "machine learning," "artificial intelligence," "chronic kidney disease," and "End-stage kidney disease" **(See S1 File)**. PubMed, Embase (OVID interface, 1947 onwards), Web of Science, and CENTRAL were chosen for their broad coverage across biomedical, nursing, allied health, and general scientific literature, while IEEE Xplore was included for coverage of more technical literature in data science. Additional articles will be retrieved by manually scrutinizing the reference lists of relevant publications.

## Study records

**Data management.**   Following database searching, studies will be populated into Covidence systematic review software [29], which will manage study selection and data extraction.

**Selection process.**   We will carry out two stages of screening. After study de-duplication through Covidence, two reviewers from a team of eight reviewers will screen the titles and abstracts of potential studies independently. We will eliminate abstracts in the initial screen if they do not report ML-based prediction models in CKD.

Included studies will undergo full-text review against the full eligibility criteria. Reasons for exclusion will be recorded for each study. Disagreement between two reviewers at each article screening and selection stage will be resolved by consensus and a third person if necessary. The PRISMA 2020 flow diagram will be generated to describe the workflow and identification of included studies for the systematic review [30].

All reviewers will receive prior training on systematic review methodology, introduction to machine learning in healthcare and the study protocol including the eligibility criteria from lead reviewers (FC, PK and KP). Each reviewer will screen with the lead reviewer (FC) for the initial 20 articles to ensure consistency. Regular team meetings will be held to resolve conflicts and ensure consistent validity across team members.

Screening Interrater reliability will be calculated in both percentage agreement and Cohen's Kappa [31].

**Data collection and management.**   The data extraction form will be designed prospectively before data collection and will be pilot tested and refined. Five members of the reviewer team who will participate in data extraction process will receive dedicated instructions and supplementary coding manual with explanations and examples of variables from the lead

reviewer (FC). During the pilot phase, the reviewer team will independently extract a random sample of five studies and review inter-rater reliability. Team training and piloting will continue until a Cohen's kappa of 0.60 (Moderate) is reached. Two independent reviewers from the reviewer team will then begin formal extraction process of the following data items based on items included in the CHARMS checklist: 1) source of data; 2) participant information; 3) outcome(s) to be predicted; 4) candidate predictors; 5) sample size; 6) Missing data; 7) Model development; 8) Model performance; 9) Model evaluation; 10) Results (final model presented) including model performance; 11) interpretation and discussion. In addition, information will be extracted on the study information (authors, year of publication, study design, journal, contact information, study period, geographical location (area and country), and funding), the assessment of reporting standards using an objective measure if mentioned in the study, and any other relevant information. All relevant text, tables, and figures will be examined for data extraction. Disagreements between the two independent reviewers will be resolved by consensus and a third reviewer if necessary. We will contact the study authors to request incompletely reported data in included studies. We will conduct analyses using available data if no response is received within 14 days.

Extraction interrater reliability will be calculated in both percentage agreement and Cohen's Kappa [31].

## Reporting quality and risk of bias

**Reporting quality assessment.**　We will assess the reporting quality of studies against the TRIPOD (Transparent reporting of a multivariable prediction model for individual prognosis or diagnosis) statement, which aims to improve the transparent reporting of prediction modeling studies in all medical settings [32].

The TRIPOD statement provides recommendations for reporting studies on developing, validating, or updating a prediction model. To assess the completeness of reporting amongst each publication, we will utilize the published "TRIPOD Adherence Extraction Form,"which evaluates 22 main items deemed essential in evaluating the transparency of prediction model studies [33]. Each article will only be assessed for items and sub-items that it applies to (development, external validation, or incremental value reporting of prediction models) based on guidance from the adherence form. Each TRIPOD item is given adherence elements to help evaluate an item. The presence or lack of an adherence element within the article will be marked down either as a "Yes" or a "No." For a TRIPOD item to receive a score, all adherence elements must be present. Overall article's TRIPOD score can be calculated by summing up adhered TRIPOD items and dividing by the total number of applicable TRIPOD items for that article. Findings on reporting quality from the TRIPOD adherence extraction will be summarised and graphically presented.

**Risk of bias assessment.**　We will assess the risk of bias (ROB) of studies by applying PROBAST (prediction model risk of bias assessment tool) [34]. PROBAST was developed to assess ROB and applicability concerns of a study that evaluates (e.g., develops or validates) a multivariable diagnostic or prognostic prediction model. Reviewers will assess each study based on the published "PROBAST Assessment Form" [35]. PROBAST is organized into four domains: participants, predictors, outcomes, and analysis. A ROB rating (high, low, or unclear) will be assigned to each domain based on answering signaling questions provided by the PROBAST assessment form. Signaling questions can be answered as yes, probably yes, no, probably no, or no information. Based on ROB results from the four domains, an overall ROB rating and prediction model applicability rating will be given to the prediction model following the recommendations in the PROBAST assessment form. A tabular presentation on PROBAST results

for each study will be available. Results will be summarized and graphically presented for each domain.

## Comments on methodology

Three reviewers from the reviewer team will assess the study quality and risk of bias. To improve review consistency, reviewers will gain thorough familiarity with the respective explanation and elaboration documents for TRIPOD and PROBAST [36, 37]. During the pilot phase, the reviewer team will independently review a random sample of five studies and review inter-rater reliability. Team training and piloting will continue until a Cohen's kappa of 0.80 (Strong) is reached. Formal assessment of study quality and risk of bias will begin by two reviewers independently. Disagreements between two reviewers will be resolved by consensus and a third reviewer.

The interrater reliability of the study quality and risk of bias assessment will be calculated in both percentage agreement and Cohen's Kappa [31].

We will contact the author if not enough information is available for assessment. We will utilize the available data if the authors do not respond for 14 days. We will present reporting quality and risk of bias assessment in the respective tables.

## Data synthesis

**Qualitative synthesis.** We will summarize and analyze the studies that meet the eligibility criteria by themes following the primary and secondary outcomes. We will report details of prediction model performance, comparing ML-based and non-ML-based models, the prediction model use case, choice of predictors and outcomes, ML model construct, prediction model reporting standard, and risk of bias. In addition, we will analyse the studies and their results following standard 4.2 –Conduct a qualitative synthesis, chapter four of Finding What Works in Health Care: Standards for Systematic Review [38], which involves the description of the clinical and methodological characteristics of individuals studies, including their strength and weaknesses, and their relevance to the particular populations and clinical settings.

**Quantitative synthesis.** The studies will likely show significant clinical, methodological and statistical heterogeneity. We will therefore synthesize data quantitatively in appropriate subgroups (prediction of CKD development, CKD diagnosis, and prediction of CKD progression to ESKD), guided by Collins et al 2022 –specific to the conduct of systematic reviews of prediction models, including the methods for quantitative synthesis [39].

**Measure of effect size.** Performance measures of the ML algorithms will be recorded, including AUROC (or c statistics), 2x2 confusion matrix, sensitivity and specificity to summarize discrimination, and observed vs. expected ratio (O:E) to summarize calibration, or any other measures utilized by individual studies.

**Assessment of heterogeneity.** We will assess the clinical heterogeneity of studies based on the ML algorithm use case, the participant characteristics, the predictor choice, and selection. We will assess the methodological heterogeneity based on the ML algorithm construct regarding the data size, ML technique, and performance measures. We will assess statistical heterogeneity using the $\chi^2$ test and the $I^2$ statistic. We will consider an $I^2$ value greater than 50% indicative of substantial heterogeneity.

**Quantitative data synthesis.** As recommended by Collins et al 2022 [39], random-effects meta-analysis will be used to summarize estimates of model discrimination and calibration.

In order to compare the performance between ML and non-ML methods, we will utilize methods described in Christodoulou et al. [40] by analyzing pairwise differences in logit AUROCs between ML-based and non-ML based techniques by random effects modeling by

DerSimonian and Laird method, either pooled or within subgroups stratified by the risk of bias, study outcomes, ML techniques. The estimate and the 95% CI will describe the pairwise differences in logit AUROCs. The average logit(AUC) difference and 95% CI pooled and stratified by the above subgroups will be calculated as an indicator of how prediction models utilizing ML compare with those utilizing non-ML methods.

The meta-analysis will be performed using Review Manager version 5.4.1 (The Cochrane Collaboration, The Nordic Cochrane Centre, Copenhagen, Denmark) [41] and Stata [42].

**Additional analysis.** Further subgroup analyses will be performed to explore possible sources of heterogeneity based on the following: study quality (risk of bias and reporting quality assessment), ML techniques, stages of validation, and dataset (stratified as determined by resulted studies).

We will conduct sensitivity analyses based on study quality, study publication years (stratified by year), study populations, the ML model construct, the ML user case, or any other relevant strata.

## Publication bias

A funnel plot will be constructed to assess the risk of publication bias.

## Confidence in cumulative evidence

Overall evidence quality will be assessed using The Grade of Recommendations of Assessment, Development, and Evaluation (GRADE) guidance for assessing strength of evidence for diagnostic tests, incorporating domains including the risk of bias, consistency, directness, precision and publication bias [43]. The overall level of evidence will be summarized into high, moderate, low and very low. The overall grade will start at "high", a serious concern/high risk of bias within one domain will result in a deduction in one level.

## Systematic review reporting

As ML in CKD is a rapidly developing field, if the number of studies that meet the eligibility criteria exceeds the capacity to report the study outcomes in one systematic review, the research team will report diagnostic and prognostic tools as separate systematic reviews to ensure a clear and focused reporting and appraisal of study outcomes.

## Discussion

Artificial intelligence and machine learning have significant potential in modern healthcare. Specifically, models developed for the prediction and diagnosis of CKD development and progression can allow for early disease recognition and intervention, which may help to facilitate early disease prevention and diagnosis, assist with early care planning, and allocate resources for the most significant clinical and economic benefit.

As the number of algorithms grows exponentially, the focus should direct toward addressing the barriers to clinical implementation. This review aims to assess the methodological rigor of model development and compare ML-based algorithms' performance with conventional methods. This will inform clinicians and technical specialists of the current development of ML in CKD care, as well as direct future model development and standardization.

## Supporting information

**S1 Checklist. PRISMA-P 2015 checklist.**
(DOCX)

**S1 File.**
(DOCX)

## Author Contributions

**Conceptualization:** Fangyue Chen, Piyawat Kantagowit, Tanawin Nopsopon, Arisa Chuklin, Krit Pongpirul.

**Methodology:** Fangyue Chen, Piyawat Kantagowit, Tanawin Nopsopon, Arisa Chuklin, Krit Pongpirul.

**Supervision:** Krit Pongpirul.

**Writing – original draft:** Fangyue Chen, Arisa Chuklin.

**Writing – review & editing:** Fangyue Chen, Piyawat Kantagowit, Tanawin Nopsopon, Krit Pongpirul.

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
