## [Decision Letter · Decision Letter 0]

16 Jan 2023

PONE-D-22-31942Prediction and diagnosis of chronic kidney disease development and progression using machine-learning: protocol for a systematic review and meta-analysis of reporting standards and model performance.PLOS ONE Dear Dr. Pongpirul,

Thank you for submitting your manuscript to PLOS ONE. After careful consideration, we feel that it has merit but does not fully meet PLOS ONE’s publication criteria as it currently stands. Therefore, we invite you to submit a revised version of the manuscript that addresses the points raised during the review process.

We look forward to receiving your revised manuscript.

Kind regards,

Zhe He, PhD

Academic Editor

PLOS ONE

Journal Requirements:

2. In your cover letter, please confirm that the research you have described in your manuscript, including participant recruitment, data collection, modification, or processing, has not started and will not start until after your paper has been accepted to the journal (assuming data need to be collected or participants recruited specifically for your study). In order to proceed with your submission, you must provide confirmation.

**Additional Editor Comments:**

Both reviewers raised a few issues to be addressed by the authors. Please submit a revised manuscript with a response to review letter detailing how these issues are addressed.

Reviewers' comments:

Reviewer's Responses to Questions

**Comments to the Author**

1. Does the manuscript provide a valid rationale for the proposed study, with clearly identified and justified research questions?

Reviewer #1: Yes

Reviewer #2: Yes

2. Is the protocol technically sound and planned in a manner that will lead to a meaningful outcome and allow testing the stated hypotheses?

Reviewer #1: Yes

Reviewer #2: Yes

3. Is the methodology feasible and described in sufficient detail to allow the work to be replicable?

Reviewer #1: Yes

Reviewer #2: Yes

4. Have the authors described where all data underlying the findings will be made available when the study is complete?

Reviewer #1: No

Reviewer #2: No

5. Is the manuscript presented in an intelligible fashion and written in standard English?

Reviewer #1: Yes

Reviewer #2: Yes

6. Review Comments to the Author

You may also provide optional suggestions and comments to authors that they might find helpful in planning their study.

Reviewer #1: The author proposed to systematically review existing machine learning methods applied to chronic kidney diseases. The authors plan to compare the quality of the literatures using multiple published rating schemes and analyzes and compare their performance using statistical and meta-analysis technique. The authors address an important question on providing a collected rating and comparison for the field and the proposed methods are sound. In addition, I have the following comments.

Are the same independent reviewers participating in all the stages of the analysis?

The instruction given to the reviewer and the calibration process at all stage should be detailed.

The interrater reliability at all stage should also be included in the final results.

A third reviewer is mentioned in the selection process but not on other stages. Will a third reviewer also involved if consensus cannot be agreed upon?

Line 115

Are studies that include teenager excluded or just with that part of the result removed?

Line122

Any plan to include studies with only conventional techniques as a comparison? I do think at least conventional methods that are widely adopted by the field should be included if not already included as part of the ML papers as a baseline model.

Line 195

The authors should include a reference to the "TRIPOD Adherence Extraction Form”

Line 211

The authors should include a reference to the "PROBAST Assessment Form”

Line 230

The sentence is confusing. The “and” is a typo and it would be better to put a quote on the whole reference. A brief description of this standard, especially how the authors are going to use it, should also be included.

Line 238

What is “the studies” referring to? If the authors mean “This study” it would be clearer to write it this way.

Line 240

Same advice as Line 230

Line 255

A citation is needed for every model and methods used in this paragraph

Line 287 and 288 need clarification.

Reviewer #2: This is a review paper, focusing on comparing machine learning based models and non-machine learning based models for prediction and diagnosis of (Chronic Kidney disease) CKD development and progression. According to the paper, this systematic review aims to answer how ML-based prediction tools in CKD development and progression perform compared with tools developed using conventional techniques.

In general, after reviewing this systemic review, I believe it is a rigorous work with clear academic goal. The paper collection follows well-defined and strict protocol, and the evaluation on each machine-learning-based model follow rigorous guidelines. As a result, I believe that the comparison outcome is meaningful and valuable. However, this paper is lack of details about the comparison result: What are the qualitative and quantitative scores of each ML-model/paper you choose? What is the method or neural network structure in each paper you collected? The author should provide this details to make this paper more convincing, and it will be a good review paper as long as these details are given. Hence, in general, I recommend minor revision to this review paper, where this "minor revision" means exactly the comparison details.

Primarily, the legitimacy of this review is proved by its registration in International Prospective Register of Systematic Reviews. Also, this systemic review is a standard work since it is prepared using the Preferred Items for Systematic Review and analysis Protocols (PRISMA-P) guidelines. Then, in order to choose related papers, the authors settle up inclusion and exclusion criterias, which focus on discovering machine-learning applications on CKD. After that, independent individuals will do data collection and management. For quality reporting, a criteria called TRIPOD (Transparent reporting of a multivariable prediction model for individual prognosis or diagnosis) is applied, while for risk of bias evaluating, a criteria called PROBAST (prediction model risk of bias assessment tool) is applied. Finally qualitative and quantitative data synthesis are conducted by independent individuals following protocols.

In general, this review paper always follow well-define protocols and process for paper review and model evaluation. Hence, after providing the details on model evaluation and model comparison, it should be accepted.

7. PLOS authors have the option to publish the peer review history of their article (what does this mean?). If published, this will include your full peer review and any attached files.

Reviewer #1: No

Reviewer #2: **Yes: **Canlin Zhang

---

## [Author Response · Author response to Decision Letter 0]

2 Feb 2023

Reviewer #1: The author proposed to systematically review existing machine learning methods applied to chronic kidney diseases. The authors plan to compare the quality of the literatures using multiple published rating schemes and analyzes and compare their performance using statistical and meta-analysis technique. The authors address an important question on providing a collected rating and comparison for the field and the proposed methods are sound. In addition, I have the following comments.

Response: We are pleased to know that our work is considered useful.

Reviewer #1: Are the same independent reviewers participating in all the stages of the analysis? The instruction given to the reviewer and the calibration process at all stage should be detailed. The interrater reliability at all stage should also be included in the final results.

Reponse: All stages of the review will be conducted by the same group of eight reviewers, during which two independent reviewers will review each stage of every study, with conflicts resolved by consensus and a third reviewer if necessary. This has been revised in the manuscript (Lines 165, 182, and 237). Details of instructions for the reviewers, the pilot and calibration process have been added to all stages of the systematic review (Lines 173-177, 183-189, and 237-244). Methods of interrater reliability assessment are also added (Line 178-179, 201-202, and 245-246).

Reviewer #1: A third reviewer is mentioned in the selection process but not on other stages. Will a third reviewer also involved if consensus cannot be agreed upon?

Response: Yes. The involvement of a third reviewer has been added for extraction, reporting quality and risk of bias assessment (Line 197-198; line 243-244).

Reviewer #1: Line 115: Are studies that include teenager excluded or just with that part of the result removed?

Response: Studies that include both adults and teenagers below above 18 years old will be excluded, given different epidemiological profile between age groups which can affect the machine learning model performance, and the difficulty in excluding teenagers from the results of individual studies, which can alter the construct of the machine learning model. An additional exclusion criterion (studies that include participants < 18 year olds) has been added to the study eligibility criteria.

Reviewer #1: Line 122: Any plan to include studies with only conventional techniques as a comparison? I do think at least conventional methods that are widely adopted by the field should be included if not already included as part of the ML papers as a baseline model.

Response: Thank you for this invaluable comment. After having conducted a search that include only machine-learning based studies, the reviewers felt that if the search criteria include studies with only conventional studies, the number of articles may exceed the capacity for study appraisal, synthesis and meta-analysis. Secondly, the reviewers felt that it may be difficult to directly compare prediction model studies that utilize conventional methods only with those utilizing machine learning methods, given the differences in the data characteristics and study participants between different studies. Instead, the comparison may be more valid within individual studies that vary only by the methods used (conventional vs. machine-learning based). After a preliminary search, the reviewers have found that most machine-learning based prediction models have been compared with conventional models widely adopted by the field (e.g. Kidney Failure Risk Equation). In addition, existing systematic reviews have been published recently (See reference 23. Ramspek, C. L., de Jong, Y., Dekker, F. W., & van Diepen, M. (2020). Towards the best kidney failure prediction tool: a systematic review and selection aid. Nephrology, dialysis, transplantation, 35(9), 1527–1538), which have comprehensively summarized available models predicting kidney failure in CKD patients. Most of the models were constructed utilizing conventional models. It will act as a useful comparator article to this proposed systematic review, which will be useful to witness how the field of machine learning has grown over the past five years.

Reviewer #1: Line 195: The authors should include a reference to the "TRIPOD Adherence Extraction Form”.

Response: The reference was added (Reference 33).

Reviewer #1: Line 211: The authors should include a reference to the "PROBAST Assessment Form”

Response: The reference was added (Reference 35).

Reviewer #1: Lines 230 and 240: The sentence is confusing. The “and” is a typo and it would be better to put a quote on the whole reference. A brief description of this standard, especially how the authors are going to use it, should also be included.

Response: Thank you. The typo has been corrected. The reviewers have added a brief description of the standard and how it will be used (Lines 258-260). We have found a recently published book chapter “Systematic Reviews of Prediction Models” that is more applicable to the study aim, and contains guidance on the conduct of quantitative synthesis (Lines 264-266).

Reviewer #1: Line 238: What is “the studies” referring to? If the authors mean “This study” it would be clearer to write it this way.

Response: “The studies” have been changed to “The studies that meet the eligibility criteria”. The authors have also placed this section (summary and analysis of primary and secondary outcomes) prior to the description of clinical and methodological characteristics of individuals studies, given it forms the main study outcomes (Line 252).

Reviewer #1: Line 255: A citation is needed for every model and methods used in this paragraph

Response: Please see the revised quantitative methods and references (Lines 280-289).

Reviewer #1: Lines 287 and 288 need clarification.

Response: Please see the rephrasing of the paragraph (Lines 304-309). We decided to use guidance from Singh S, Chang SM, Matchar DB, Bass EB. Chapter 7: grading a body of evidence on diagnostic tests. J Gen Intern Med. 2012;27 Suppl 1(Suppl 1):S47-S55., as ML-based prediction tools are more akin to diagnostic tests. The body of evidence can thus be assessed in a similar way.

Reviewer #2: This is a review paper, focusing on comparing machine learning based models and non-machine learning based models for prediction and diagnosis of (Chronic Kidney disease) CKD development and progression. According to the paper, this systematic review aims to answer how ML-based prediction tools in CKD development and progression perform compared with tools developed using conventional techniques.

In general, after reviewing this systemic review, I believe it is a rigorous work with clear academic goal. The paper collection follows well-defined and strict protocol, and the evaluation on each machine-learning-based model follow rigorous guidelines. As a result, I believe that the comparison outcome is meaningful and valuable. However, this paper is lack of details about the comparison result: What are the qualitative and quantitative scores of each ML-model/paper you choose? What is the method or neural network structure in each paper you collected? The author should provide this details to make this paper more convincing, and it will be a good review paper as long as these details are given. Hence, in general, I recommend minor revision to this review paper, where this "minor revision" means exactly the comparison details.

Primarily, the legitimacy of this review is proved by its registration in International Prospective Register of Systematic Reviews. Also, this systemic review is a standard work since it is prepared using the Preferred Items for Systematic Review and analysis Protocols (PRISMA-P) guidelines. Then, in order to choose related papers, the authors settle up inclusion and exclusion criterias, which focus on discovering machine-learning applications on CKD. After that, independent individuals will do data collection and management. For quality reporting, a criteria called TRIPOD (Transparent reporting of a multivariable prediction model for individual prognosis or diagnosis) is applied, while for risk of bias evaluating, a criteria called PROBAST (prediction model risk of bias assessment tool) is applied. Finally qualitative and quantitative data synthesis are conducted by independent individuals following protocols.

In general, this review paper always follow well-define protocols and process for paper review and model evaluation. Hence, after providing the details on model evaluation and model comparison, it should be accepted.

Response: We thank the reviewer for the compliments, comments, and suggestions. We noted the recent publication of Systematic Reviews in Health Research: Meta-Analysis in Context, Chapter 18 Systematic Reviews of Prediction Models (Ref 39).

The principal outcome that the article focuses on is the comparison in the performance of machine-learning based models with the conventional-methods based models. The book chapter has recommended the use of performance measures for both calibration and discrimination, the most common being c-statistics (AUROC) and O : E. We have therefore decided to utilize AUROC as the main comparison measure, given it is previously used in several other systematic reviews of prediction models (Ref 21, 40).

The average logit (AUC) difference and 95% CI, pooled and stratified by subgroups (risk of bias, study outcomes, ML techniques) will be calculated, and will be the main quantitative outcome to allow for the comparison between the performance of ML vs. non-ML methods.

---

## [Editor Report · Decision Letter 1]

9 Feb 2023

Prediction and diagnosis of chronic kidney disease development and progression using machine-learning: protocol for a systematic review and meta-analysis of reporting standards and model performance.

PONE-D-22-31942R1

Dear Dr. Pongpirul,

We’re pleased to inform you that your manuscript has been judged scientifically suitable for publication and will be formally accepted for publication once it meets all outstanding technical requirements.

Kind regards,

Zhe He, PhD

Academic Editor

PLOS ONE

Additional Editor Comments (optional):

The authors have addressed all the comments from the reviewers.

---

## [Editor Report · Acceptance letter]

14 Feb 2023

PONE-D-22-31942R1 

Prediction and diagnosis of chronic kidney disease development and progression using machine-learning: protocol for a systematic review and meta-analysis of reporting standards and model performance 

Dear Dr. Pongpirul:

I'm pleased to inform you that your manuscript has been deemed suitable for publication in PLOS ONE. Congratulations! Your manuscript is now with our production department. 

Kind regards, 

on behalf of

Dr. Zhe He 

Academic Editor

PLOS ONE